# Correcting Class Imbalance in Prior-Data Fitted Networks for Tabular Classification

## Abstract

Prior-data fitted networks (PFNs) have achieved exceptional performance on tabular classification tasks. However, like other classifiers, their performance can suffer under the effect of class imbalance, resulting in poor performance for rare classes. Several techniques exist which attempt to mitigate the deleterious effect of class imbalance on classification performance, but the in-context learning (ICL) dynamic of PFNs means that loss-based strategies are impossible, and other techniques are unproven. We have adapted several classical techniques addressing class imbalance and analyzed their performance on PFN classification. We observe that thresholding performs exceptionally well because of the calibration characteristics of PFNs, and downsampling performs comparably because of PFNs exceptional limited-data performance, with the additional benefit of reduced computation cost for inference.

## 1. Introduction

Prior-data fitted networks (PFNs) have become ubiquitous in critical areas such as tabular data inference (Grinsztajn et al., 2025) and timeseries forecasting (Onur Taga et al., 2025). Specifically, TabPFN (and its variants) has demonstrated extraordinary performance on tabular classification and regression tasks *without updating any weights* and with very little task-specific training data. PFNs achieve this outstanding efficiency by pretraining on copious amounts of *synthetic* data specifically for the task of in-context learning.

By training on synthetic data from only highly structured causal models, PFNs implicitly learn to estimate the posterior predictive distribution (PPD) over a prior determined by the class of data generating models. The relationship between the pretraining model class and the final prediction

is critical, as the predictions of the classifier will follow the structure of the data generating models used in training. In this way, controlling the pretraining prior controls the hypothesis space of the classification.

Another key architectural aspect of PFNs is their reliance on in-context learning (ICL). ICL is a form of meta-learning where the PFN model is trained not to predict outputs for a fixed task, but to predict the relationship between a labeled context set and an unlabeled query. This results in a model whose weights are not updated at all for a new task; instead, the prediction relies on a relatively few labeled examples (context) given to the model. This is distinct from classical in-weight learning where the task-specific examples are used to update model weights before passing in the query to predict a label. ICL is typically performed using transformer models, so the relationships between context and query is captured using the attention mechanism.

While PFNs achieve state of the art performance on tabular classification, like most classification models, they perform poorly on class-imbalanced data, where some classes are vastly more prevalent than others. In fact, these models may achieve reasonable or even stellar average accuracy, but the performance on the minority classes can be significantly degraded due to limited samples. This, in turn, limits detection of rare classes (e.g., rare disease or cyberattack detection).

Addressing the effects of class imbalance falls in three major categories: loss-level, data-level, or decision-level. Loss reweighting is a classical method that falls in the first category, and involves upweighting the loss on minority class samples such that they have the same influence on the model as the majority. Data-level methods include downsampling the majority size to match that of the minority as well as generating synthetic minority samples. Decision-level methods involve manipulating the output of the model, such as scaling/tilting the soft score from the classifier. All of these methods have merit; however, since the learning dynamics of ICL are distinct from in-weight learning, some of these techniques, in particular loss reweighting, cannot be applied to this setting.

Recently, (Ma et al., 2024) addressed the performance of TabPFNs on rare classes by evaluating the efficacy of generating synthetic minority samples. This technique is limited

[1]Anonymous Institution, Anonymous City, Anonymous Region, Anonymous Country. Correspondence to: Anonymous Author <anon.email@domain.com>.

Preliminary work. Under review by the International Conference on Machine Learning (ICML). Do not distribute.

by the effectiveness of the method used to generate the synthetic samples, as any distortion of the distribution of the synthetic samples will affect the downstream classification. It is also computationally expensive, since a sample must be generated for each additional sample in overrepresented classes.

Our preliminary investigation of the unique calibration characteristics of PFNs motivates this work: a comparison of a variety of well-studied correction methods including thresholding, downsampling, oversampling, and synthetic upsampling (all defined formally in Section 2.3). For binary classification tasks, our results show that thresholding achieves the best performance, with a drastic increase in minority class performance with a minimal decrease in majority performance. Downsampling also performs well, achieving the highest worst-class accuracy with only a slight decrease in balanced performance and the additional benefit of decreasing inference computation cost by reducing the number of context samples.

## 2. Problem Setup

### 2.1. Prior-Data Fitted Networks (PFNs)

PFNs are a model class trained on a Bayesian prior over supervised learning tasks, typically with large-scale synthetic datasets, so that they can use in-context learning (ICL) to predict the posterior predictive distribution (PPD) directly (Müller et al., 2021). During training, these models are given a set of context points drawn from a distribution and trained to predict the masked label of a query point drawn from the same distribution. Minimizing a standard cross entropy loss over a variety of distributions leads the model to learn $P(y \mid x, D)$, the PPD of the context $D = \{x_c, y_c\}_{i=1}^n$ and query $x$. Formally, the PPD can be written as (Hollmann et al., 2022):

$$P(y|x, D) \propto \int_\Phi P(y|x, \phi)P(D|\phi)P(\phi)d\phi, \quad (1)$$

where $y$ is the class label of data point $x$, and $D$ is a labeled dataset drawn from the same distribution as $x$, and $\Phi$ is the set of data generating functions.

The set of data generating functions is a prior which defines the hypothesis space of the classifier. A common selection is the set of structural causal models (SCMs), where the causal relationships between features are represented by the edges of a directed acyclic graph (Hollmann et al., 2022). This trains the model to estimate the PPD directly by *implicitly* predicting the SCM which best fits the data and using that model to predict $\hat{y}$.

### 2.2. In-Context Learning (ICL)

During ICL, the pretrained model is given a set of $n$ labeled context points, $\{(x_c, y_c)\}_{i=1}^n$, and a query, $x_q$, and is asked to predict $\hat{y}_q$, the label of the query point. Critically, this does not involve updating model weights and is thus distinct from standard in-weight learning, where labeled data is used to update model weights before passing in the query to predict a label.

The idea of ICL originated with large language models (LLMs), when their ability to perform task-agnostic few-shot classification was first demonstrated (Brown et al., 2020). However, since then, the idea of ICL has expanded, and models have been trained to perform ICL explicitly in a variety of settings, including images (Wang et al., 2023) and tabular data (Hollmann et al., 2022).

The learning of an ICL model takes place entirely within the latent representation of the context and query data rather than the weights of the model, which can make manipulating the model difficult. Without performing a computationally expensive fine-tuning operation on the model, the only ways to affect its performance are to manipulate the input data or the downstream prediction.

### 2.3. Data-Level Strategies (Sampling)

**Downsampling** involves removing majority samples from the context set, such that the number of samples from each class becomes equal. This achieves $\pi_0 = \pi_1$ at the cost of decreasing the available information about the majority class.

**Oversampling** involves including samples from the minority class in the context set multiple times to equalize the number of samples from each class. This technique also achieves $\pi_0 = \pi_1$, but distorts the minority distribution, making it appear spikier than the true distribution.

**Synthetic Upsampling** involves generating artificial samples of the minority class using the context set and supplementing the context set with enough of these samples that it becomes class-balanced. Synthetic upsampling is similar to oversampling, but the distortion of the minority distribution comes from any inaccuracies in the distribution learned by the generator.

### 2.4. Decision-Level Strategies

The Bayes optimal rule for classification picks the class with the highest soft-score. For binary classification, this simplifies to setting a threshold of 0.5 to make a hard decision.

**Thresholding** involves moving the decision boundary away from 0.5. By doing so, we can counteract the effect of any consistent bias towards the majority class from learning

from an imbalanced context set, resulting in an increase in downstream balanced prediction.

## 2.5. Metrics

Our metrics of interest are the per-class, balanced, and worst-class accuracy (WCA). We use a hold-out test set to compute accuracies. Per-class accuracy is simply the fraction of correctly classified points in each class of the test data, i.e., $P(\hat{y} = i | y = i), i \in \{0, 1\}$. The average test accuracy is simply the empirical accuracy over both classes, i.e., each accuracy is scaled by its empirical prior. On the other hand, balanced accuracy is the average of the two per-class accuracies (equivalently, the average accuracy on a balanced test set). Finally, WCA is the minimum of the two per-class accuracies.

## 3. Experimental Results & Analysis

For our experiments, we select a subset of the benchmark collection of datasets, OpenML-CC18. For details on selection criteria, see Appendix A. For the model, we utilize TabPFN-2.5 (Grinsztajn et al., 2025) which offers state-of-the-art performance among tabular PFN models.

### 3.1. Calibration

In Figure 1, we illustrate the empirically computed calibration of TabPFN over several datasets when the training context is either balanced or imbalanced. We observe a persistent trend of good calibration with balanced datasets and majority class bias in the imbalanced setting. Figure 1 shows the calibration curve averaged over all of the datasets and three context sizes. In the balanced setting ($\pi_1 = 0.5$), we see that the model is well-calibrated, but as the data becomes imbalanced, the model becomes increasingly majority-biased. In general, most non-linear classifier models are not well-calibrated and are either overconfident or underconfident for all predicted probabilities, and are often addressed with tilting or scaling methods (Niculescu-Mizil & Caruana, 2005). In contrast, TabPFNs are simply biased towards the majority class for which the solution can be simple thresholding. This is a critical observation which, to the best of our knowledge, has not been exploited in the PFN literature.

### 3.2. Thresholding

Based on the calibration observations above, it is natural to ask how sensitive the per-class accuracies are to the choice of threshold for a given dataset. To evaluate this effect, we evaluate TabPFN performance with a variety of decision thresholds, $\tau$. When we do this, we observe that the maximum balanced accuracy appears approximately at the expected value of $\tau = \pi_1$.

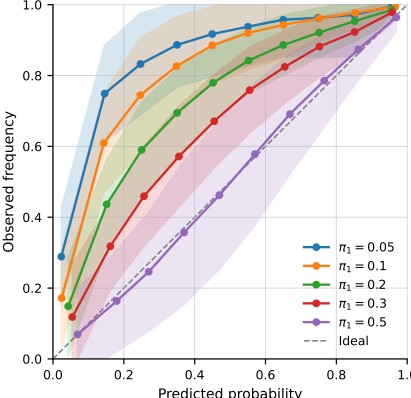

*Figure 1.* Calibration Curve Averaged Over Datasets and $N \in \{100, 500, 1000\}$

Figure 2 shows an example of one such experiment on the kr-vs-kp dataset with a context size of $N = 500$ samples and an imbalance of $\pi_1 \in \{0.05, 0.1, 0.2, 0.5\}$. We see that even as the imbalance changes and the crossover point moves, the maximum balanced accuracy point tracks $\tau = \pi_1$.

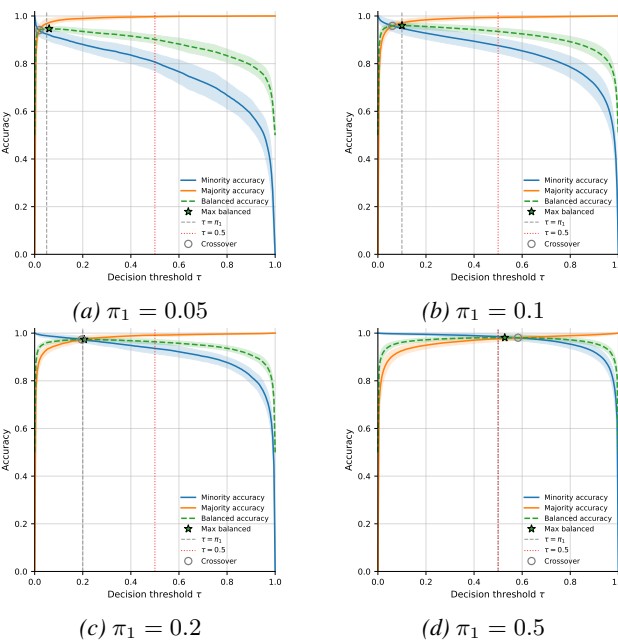

*(a) $\pi_1 = 0.05$*  *(b) $\pi_1 = 0.1$*

*(c) $\pi_1 = 0.2$*  *(d) $\pi_1 = 0.5$*

*Figure 2.* Threshold crossover for kr-vs-kp with $N = 500$ across imbalance levels $\pi_1 \in \{0.05, 0.1, 0.2, 0.5\}$

This result validates our decision to use a threshold of $\tau = \pi_1$ as a correction method to attempt to maximize balanced accuracy. We remark that the idea of thresholding is very standard in detection theory; however, in modern learning-based classification, average accuracy (averaged over priors) is often the only metric presented which results in these models ignoring the performance on rare classes, especially

when the average accuracy is high. Our work shows that the performance of SOTA models like TabPFN can be enhanced by judicious use of thresholding and downsampling.

### 3.3. Downsampling

While thresholding attempts to adjust the output of the model to account for a difference in the priors, downsampling adjusts those priors directly by reducing the number of majority samples. To investigate the effect of different levels of downsampling on classification performance, we fix a number of class 1 (minority) samples and sweep the number of class 0 (majority) samples. Note that we still denote class 0 as the majority, even when it is sampled down such that $\pi_0 < \pi_1$.

In these experiments, we see that the balanced accuracy is approximately constant while the number of class 0 samples is greater than or equal to the number of class 1 samples, though as the number of class 0 samples increases the worst-class accuracy decreases. Figure 3 shows an example of this evaluation for `kr-vs-kp` with $N_{min} = 50$.

This justifies the decision to downsample to $\pi_0 = \pi_1$. Downsampling to the lowest level before the rapid falloff in balanced accuracy has the additional benefit of noticeably reducing required computation, since TabPFN query computation scales quadratically with context size (Hollmann et al., 2022).

### 3.4. Comparison of Methods

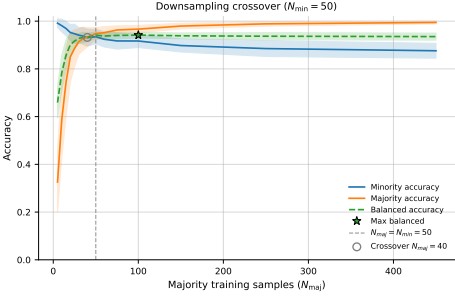

*Figure 3.* TabPFN2.5 Downsample Crossover

Figure 4 shows the average performance change (relative to no correction) across imbalance levels for each metric and correction method. For the absolute accuracies of each experiment and average rank of each method, see Appendix B.

In Figure 4, we see that thresholding and downsampling both increase minority accuracy significantly at the cost of a slight decrease in majority accuracy. Downsampling results in a slightly greater increase in minority class accuracy, but a much greater decrease in majority accuracy, leading thresholding to have a slightly higher balanced accuracy. Additionally, thresholding and downsampling perform

similarly on WCA.

We consistently see that thresholding achieves both competitive balanced and worst-class accuracy, with downsampling close behind. Meanwhile, both OS and TabPFGen perform worse than the base model alone. For OS, this can be attributed to the PFN fitting to the repeated samples resulting in a spiky posterior fit. On the other hand, it should not be surprising that the synthetic minority points generated by TabPFGen would not improve the minority accuracy of the TabPFN classifier since they are from the same model class and are generated using the same data.

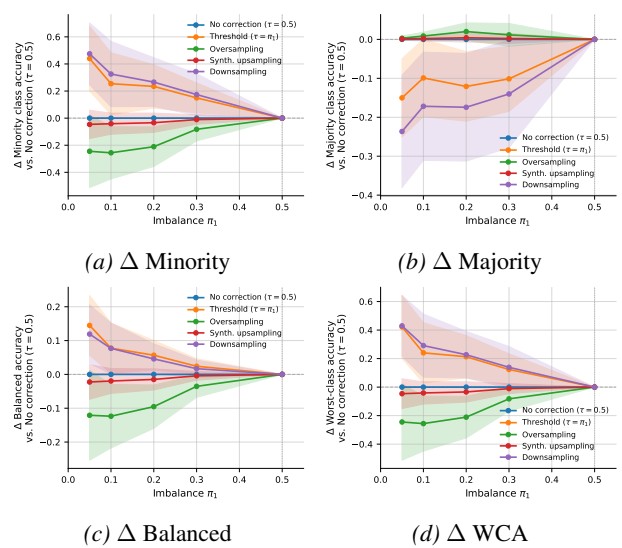

*(a) Δ Minority*      *(b) Δ Majority*

*(c) Δ Balanced*      *(d) Δ WCA*

*Figure 4.* Change ($\Delta$ *relative to no correction*) in minority (4a), majority (4b), balanced (4c), and worst-class accuracy (4d) averaged over datasets and $N \in \{100, 500, 1000\}$

## 4. Conclusion

Via a thorough empirical investigation, we have shown that thresholding and downsampling can be effectively used to reduce the negative effects of data imbalance on balanced and worst-class accuracy when performing tabular classification with PFNs. The methods which perform the best are simple and easy to implement, meaning practitioners will be able to effectively address imbalance and achieve high accuracy on classes with limited data without the addition of significant complexity. However, this simplicity may also leave some performance to be gained with more complex methods. Future work will extend this framework to multi-class classification, which will require non-trivial adaptations of the correction strategies explored here. More broadly, we hope this work opens a productive line of inquiry into principled methods for classification bias correction.

## References

Brown, T., Mann, B., Ryder, N., Subbiah, M., Kaplan, J. D., Dhariwal, P., Neelakantan, A., Shyam, P., Sastry, G., Askell, A., et al. Language models are few-shot learners. *Advances in neural information processing systems*, 33: 1877–1901, 2020.

Grinsztajn, L., Flöge, K., Key, O., Birkel, F., Jund, P., Roof, B., Jäger, B., Safaric, D., Alessi, S., Hayler, A., et al. Tabpfn-2.5: Advancing the state of the art in tabular foundation models. *arXiv preprint arXiv:2511.08667*, 2025.

Hollmann, N., Müller, S., Eggensperger, K., and Hutter, F. Tabpfn: A transformer that solves small tabular classification problems in a second. *arXiv preprint arXiv:2207.01848*, 2022.

Ma, J., Dankar, A., Stein, G., Yu, G., and Caterini, A. Tabpfgen–tabular data generation with tabpfn. *arXiv preprint arXiv:2406.05216*, 2024.

Müller, S., Hollmann, N., Arango, S. P., Grabocka, J., and Hutter, F. Transformers can do bayesian inference. *arXiv preprint arXiv:2112.10510*, 2021.

Niculescu-Mizil, A. and Caruana, R. Predicting good probabilities with supervised learning. In *Proceedings of the 22nd international conference on Machine learning*, ICML '05, pp. 625–632, New York, NY, USA, August 2005. Association for Computing Machinery. ISBN 9781595931801. doi: 10.1145/1102351. 1102430. URL https://dl.acm.org/doi/10. 1145/1102351.1102430.

Onur Taga, E., Emrullah Ildiz, M., and Oymak, S. Timepfn: Effective multivariate time series forecasting with synthetic data. *arXiv e-prints*, pp. arXiv–2502, 2025.

Wang, X., Wang, W., Cao, Y., Shen, C., and Huang, T. Images speak in images: A generalist painter for in-context visual learning. In *Proceedings of the IEEE/CVF Conference on Computer Vision and Pattern Recognition*, pp. 6830–6839, 2023.

## A. Dataset Selection

For our experiments, we select binary classification tasks from the benchmark collection of datasets, OpenML-CC18. This collection has 72 tabular classification tasks from which we select 11 datasets satisfying the following constraints:

- **Task**: binary classification

- **Test Size**: 500 examples per class

- **Train Size**: 500 minority examples and 950 majority examples (allowing imbalances of up to $\pi_1 = 0.05$ while maintaining 1000 total training examples)

The query (test) set is always balanced, and thus, the average test accuracy can also be viewed as the balanced accuracy. Table 1 shows the selected datasets along with their size and natural imbalance.

*Table 1.* Summary of the datasets used in experiments.

| Dataset | $N$ | $\pi_1$ |
|---|---|---|
| kr-vs-kp | 3,196 | 0.478 |
| spambase | 4,601 | 0.394 |
| electricity | 45,312 | 0.424 |
| jm1 | 10,885 | 0.194 |
| adult | 48,842 | 0.239 |
| Bioresponse | 3,751 | 0.458 |
| phoneme | 5,404 | 0.293 |
| nomao | 34,465 | 0.286 |
| PhishingWebsites | 11,055 | 0.443 |
| bank-marketing | 45,211 | 0.117 |
| numerai28.6 | 96,320 | 0.495 |

## B. Extended Metrics

Table 2 reports the average rank of each method across datasets and instantiations, where a method's rank in each trial is its relative position in accuracy (1 is the best). We see that thresholding has the highest average rank in most experiments, while downsampling performs the best in some others and otherwise ranks around number two.

Table 3 shows the results for each dataset, context size, imbalance level, and correction method. We report minority, majority, balance and worst-class accuracies. Note that WCA might be worse than the minimum of the minority and majority if the worst class varied by iteration.

*Table 2.* Correction Method Average Rank Over Datasets and Instantiations

| $N$ | $\pi_1$ | None | | Thrsh. | | OS | | TabPFGen | | DS | |
|---|---|---|---|---|---|---|---|---|---|---|---|
| | | Bal. | WCA | Bal. | WCA | Bal. | WCA | Bal. | WCA | Bal. | WCA |
| 100 | 0.05 | 3.27 | 3.49 | **1.35** | **1.37** | 4.40 | 4.52 | 3.78 | 3.89 | 2.20 | 1.73 |
| | 0.1 | 3.23 | 3.43 | **1.35** | **1.35** | 4.67 | 4.68 | 3.59 | 3.80 | 2.15 | 1.73 |
| | 0.2 | 3.10 | 3.30 | **1.39** | **1.35** | 4.71 | 4.76 | 3.66 | 3.82 | 2.14 | 1.76 |
| | 0.3 | 3.17 | 3.49 | **1.81** | 1.66 | 4.36 | 4.55 | 3.51 | 3.68 | 2.15 | **1.62** |
| 500 | 0.05 | 3.43 | 3.58 | **1.28** | **1.43** | 4.65 | 4.65 | 3.60 | 3.76 | 2.04 | 1.58 |
| | 0.1 | 3.39 | 3.47 | **1.32** | **1.37** | 4.73 | 4.77 | 3.66 | 3.73 | 1.89 | 1.66 |
| | 0.2 | 3.19 | 3.39 | **1.42** | **1.40** | 4.79 | 4.78 | 3.54 | 3.73 | 2.06 | 1.69 |
| | 0.3 | 3.11 | 3.40 | **1.55** | **1.53** | 4.67 | 4.64 | 3.44 | 3.69 | 2.22 | 1.75 |
| 1000 | 0.05 | 3.46 | 3.52 | 2.64 | 2.70 | 3.99 | 3.99 | 3.55 | 3.59 | **1.36** | **1.20** |
| | 0.1 | 3.39 | 3.47 | 2.64 | 2.66 | 3.98 | 4.00 | 3.53 | 3.62 | **1.46** | **1.25** |
| | 0.2 | 3.12 | 3.29 | **1.30** | **1.45** | 4.81 | 4.78 | 3.65 | 3.77 | 2.11 | 1.72 |
| | 0.3 | 3.10 | 3.30 | **1.50** | **1.49** | 4.59 | 4.61 | 3.52 | 3.75 | 2.29 | 1.85 |

*Table 3.* Results by Dataset and Correction Method across varying sample sizes ($N$) and class imbalances ($\pi_1$)

*(a)* $N = 100$ and $\pi_1 = 0.05$

| Dataset | None | | | | Thrsh. | | | | OS | | | | TabPFGen | | | | DS | | | |
|---|---|---|---|---|---|---|---|---|---|---|---|---|---|---|---|---|---|---|---|---|
| | Min. | Maj. | Bal. | WCA | Min. | Maj. | Bal. | WCA | Min. | Maj. | Bal. | WCA | Min. | Maj. | Bal. | WCA | Min. | Maj. | Bal. | WCA |
| kr-vs-kp | 0.559 | 0.994 | 0.777 | 0.559 | **0.795** | 0.916 | **0.855** | **0.794** | 0.028 | **0.999** | 0.514 | 0.028 | 0.293 | 0.995 | 0.644 | 0.293 | 0.675 | 0.651 | 0.663 | 0.546 |
| spambase | 0.421 | 0.988 | 0.705 | 0.421 | **0.815** | 0.901 | **0.858** | **0.808** | 0.045 | **1.000** | 0.523 | 0.045 | 0.084 | 1.000 | 0.542 | 0.084 | 0.716 | 0.855 | 0.785 | 0.646 |
| electricity | 0.064 | 0.997 | 0.531 | 0.064 | **0.624** | 0.774 | **0.699** | **0.592** | 0.010 | **1.000** | 0.505 | 0.010 | 0.062 | 0.997 | 0.529 | 0.062 | 0.594 | 0.628 | 0.611 | 0.545 |
| jm1 | 0.008 | 0.995 | 0.502 | 0.008 | **0.528** | 0.665 | **0.596** | **0.528** | 0.014 | 0.996 | 0.505 | 0.014 | 0.010 | **0.997** | 0.504 | 0.010 | 0.523 | 0.613 | 0.568 | 0.404 |
| adult | 0.039 | 0.999 | 0.519 | 0.039 | 0.739 | 0.725 | **0.732** | 0.630 | 0.006 | **0.999** | 0.503 | 0.006 | 0.042 | 0.998 | 0.520 | 0.042 | **0.803** | 0.647 | 0.725 | **0.647** |
| Bioresponse | 0.017 | 0.997 | 0.507 | 0.017 | 0.507 | 0.674 | **0.590** | 0.462 | 0.001 | **1.000** | 0.500 | 0.001 | 0.009 | 0.998 | 0.504 | 0.009 | **0.592** | 0.485 | 0.538 | 0.449 |
| phoneme | 0.036 | 0.992 | 0.514 | 0.036 | **0.731** | 0.715 | **0.723** | **0.647** | 0.008 | **0.998** | 0.503 | 0.008 | 0.020 | 0.997 | 0.509 | 0.020 | 0.667 | 0.729 | 0.698 | 0.640 |
| nomao | 0.415 | 0.989 | 0.702 | 0.415 | 0.809 | 0.868 | **0.838** | **0.781** | 0.102 | **1.000** | 0.551 | 0.102 | 0.165 | 1.000 | 0.582 | 0.165 | **0.826** | 0.815 | 0.821 | 0.771 |
| PhishingWebsites | 0.575 | 0.994 | 0.785 | 0.575 | 0.807 | 0.950 | **0.878** | **0.807** | 0.149 | **0.999** | 0.574 | 0.149 | 0.316 | 0.998 | 0.657 | 0.316 | 0.815 | 0.863 | 0.839 | 0.797 |
| bank-marketing | 0.086 | 0.994 | 0.540 | 0.086 | **0.656** | 0.794 | **0.725** | **0.647** | 0.001 | **1.000** | 0.500 | 0.001 | 0.025 | 0.998 | 0.512 | 0.025 | 0.612 | 0.675 | 0.644 | 0.531 |
| numerai28.6 | 0.002 | 0.998 | 0.500 | 0.002 | 0.418 | 0.595 | 0.506 | 0.396 | 0.000 | **1.000** | 0.500 | 0.000 | 0.001 | 1.000 | 0.500 | 0.001 | **0.477** | 0.537 | **0.507** | **0.443** |
| Average | 0.202 | 0.994 | 0.598 | 0.202 | **0.675** | 0.780 | **0.727** | **0.645** | 0.033 | **0.999** | 0.516 | 0.033 | 0.093 | 0.998 | 0.546 | 0.093 | 0.664 | 0.682 | 0.673 | 0.584 |

*(b)* $N = 100$ and $\pi_1 = 0.10$

| Dataset | None | | | | Thrsh. | | | | OS | | | | TabPFGen | | | | DS | | | |
|---|---|---|---|---|---|---|---|---|---|---|---|---|---|---|---|---|---|---|---|---|
| | Min. | Maj. | Bal. | WCA | Min. | Maj. | Bal. | WCA | Min. | Maj. | Bal. | WCA | Min. | Maj. | Bal. | WCA | Min. | Maj. | Bal. | WCA |
| kr-vs-kp | 0.625 | 0.988 | 0.806 | 0.625 | **0.887** | 0.917 | **0.902** | **0.866** | 0.106 | **1.000** | 0.553 | 0.106 | 0.478 | 0.995 | 0.736 | 0.478 | 0.723 | 0.713 | 0.718 | 0.643 |
| spambase | 0.576 | 0.981 | 0.778 | 0.576 | **0.863** | 0.894 | **0.878** | **0.850** | 0.130 | 0.997 | 0.564 | 0.130 | 0.280 | 0.996 | 0.638 | 0.280 | 0.819 | 0.839 | 0.829 | 0.767 |
| electricity | 0.200 | 0.991 | 0.596 | 0.200 | 0.607 | 0.835 | **0.721** | 0.607 | 0.031 | **0.998** | 0.515 | 0.031 | 0.155 | 0.993 | 0.574 | 0.155 | **0.624** | 0.737 | 0.681 | **0.615** |
| jm1 | 0.008 | **0.998** | 0.503 | 0.008 | **0.542** | 0.705 | **0.623** | **0.538** | 0.019 | 0.985 | 0.502 | 0.019 | 0.017 | 0.993 | 0.505 | 0.017 | 0.539 | 0.662 | 0.601 | 0.529 |
| adult | 0.101 | 0.986 | 0.543 | 0.101 | **0.758** | 0.737 | **0.747** | **0.698** | 0.019 | 0.996 | 0.508 | 0.019 | 0.136 | 0.978 | 0.557 | 0.136 | 0.731 | 0.724 | 0.728 | 0.654 |
| Bioresponse | 0.028 | 0.994 | 0.511 | 0.028 | 0.591 | 0.651 | **0.621** | 0.550 | 0.004 | **0.999** | 0.501 | 0.004 | 0.019 | 0.996 | 0.508 | 0.019 | **0.607** | 0.545 | 0.576 | 0.524 |
| phoneme | 0.102 | 0.987 | 0.544 | 0.102 | **0.733** | 0.763 | **0.748** | **0.710** | 0.034 | **0.997** | 0.515 | 0.034 | 0.128 | 0.988 | 0.558 | 0.128 | 0.714 | 0.732 | 0.723 | 0.665 |
| nomao | 0.642 | 0.980 | 0.811 | 0.642 | **0.875** | 0.903 | **0.889** | **0.863** | 0.261 | **0.999** | 0.630 | 0.261 | 0.376 | 0.996 | 0.686 | 0.376 | 0.836 | 0.865 | 0.850 | 0.804 |
| PhishingWebsites | 0.667 | 0.994 | 0.831 | 0.667 | **0.871** | 0.927 | **0.899** | **0.863** | 0.290 | **0.999** | 0.645 | 0.290 | 0.587 | 0.995 | 0.791 | 0.587 | 0.847 | 0.911 | 0.879 | 0.835 |
| bank-marketing | 0.150 | 0.982 | 0.566 | 0.150 | **0.695** | 0.795 | **0.745** | **0.679** | 0.014 | **0.998** | 0.506 | 0.014 | 0.088 | 0.992 | 0.540 | 0.088 | 0.677 | 0.695 | 0.686 | 0.626 |
| numerai28.6 | 0.001 | 0.998 | 0.500 | 0.001 | 0.405 | 0.593 | 0.499 | 0.405 | 0.000 | **1.000** | 0.500 | 0.000 | 0.000 | 1.000 | 0.500 | 0.000 | **0.508** | 0.494 | **0.501** | **0.452** |
| Average | 0.282 | 0.989 | 0.635 | 0.282 | **0.712** | 0.793 | **0.752** | **0.694** | 0.083 | **0.997** | 0.540 | 0.083 | 0.206 | 0.993 | 0.599 | 0.206 | 0.693 | 0.720 | 0.706 | 0.647 |

*(c)* $N = 100$ and $\pi_1 = 0.20$

| Dataset | None | | | | Thrsh. | | | | OS | | | | TabPFGen | | | | DS | | | |
|---|---|---|---|---|---|---|---|---|---|---|---|---|---|---|---|---|---|---|---|---|
| | Min. | Maj. | Bal. | WCA | Min. | Maj. | Bal. | WCA | Min. | Maj. | Bal. | WCA | Min. | Maj. | Bal. | WCA | Min. | Maj. | Bal. | WCA |
| kr-vs-kp | 0.767 | 0.986 | 0.876 | 0.767 | 0.845 | 0.968 | **0.906** | 0.845 | 0.576 | **0.997** | 0.786 | 0.576 | 0.724 | 0.990 | 0.857 | 0.724 | **0.862** | 0.870 | 0.866 | 0.809 |
| spambase | 0.784 | 0.975 | 0.879 | 0.784 | **0.890** | 0.918 | **0.904** | **0.883** | 0.506 | **0.993** | 0.749 | 0.506 | 0.667 | 0.986 | 0.826 | 0.667 | 0.839 | 0.899 | 0.869 | 0.837 |
| electricity | 0.334 | 0.961 | 0.647 | 0.334 | 0.659 | 0.803 | **0.731** | 0.658 | 0.187 | **0.979** | 0.583 | 0.187 | 0.366 | 0.955 | 0.661 | 0.366 | **0.669** | 0.734 | 0.701 | 0.630 |
| jm1 | 0.113 | 0.962 | 0.537 | 0.113 | 0.526 | 0.714 | **0.620** | 0.518 | 0.080 | **0.967** | 0.524 | 0.080 | 0.134 | 0.954 | 0.544 | 0.134 | **0.591** | 0.627 | 0.609 | **0.560** |
| adult | 0.405 | 0.947 | 0.676 | 0.405 | 0.788 | 0.772 | **0.780** | **0.743** | 0.158 | **0.980** | 0.569 | 0.158 | 0.390 | 0.949 | 0.670 | 0.390 | **0.826** | 0.681 | 0.754 | 0.663 |
| Bioresponse | 0.126 | 0.977 | 0.551 | 0.126 | **0.694** | 0.679 | **0.687** | 0.634 | 0.038 | **0.994** | 0.516 | 0.038 | 0.105 | 0.977 | 0.541 | 0.105 | 0.617 | 0.615 | 0.616 | 0.548 |
| phoneme | 0.371 | 0.945 | 0.658 | 0.371 | **0.812** | 0.739 | **0.776** | **0.726** | 0.191 | **0.971** | 0.581 | 0.191 | 0.311 | 0.949 | 0.630 | 0.311 | 0.775 | 0.715 | 0.745 | 0.686 |
| nomao | 0.778 | 0.958 | 0.868 | 0.778 | 0.884 | 0.894 | 0.889 | **0.859** | 0.536 | **0.989** | 0.762 | 0.536 | 0.645 | 0.980 | 0.812 | 0.645 | **0.901** | 0.879 | **0.890** | 0.845 |
| PhishingWebsites | 0.783 | 0.973 | 0.878 | 0.783 | **0.870** | 0.940 | **0.905** | **0.870** | 0.668 | **0.992** | 0.830 | 0.668 | 0.764 | 0.979 | 0.871 | 0.764 | 0.856 | 0.928 | 0.892 | 0.851 |
| bank-marketing | 0.361 | 0.960 | 0.661 | 0.361 | **0.752** | 0.788 | **0.770** | **0.733** | 0.056 | **0.993** | 0.524 | 0.056 | 0.275 | 0.959 | 0.617 | 0.275 | 0.748 | 0.758 | 0.753 | 0.689 |
| numerai28.6 | 0.000 | **1.000** | 0.500 | 0.000 | 0.378 | 0.625 | 0.502 | 0.378 | 0.002 | 0.999 | 0.501 | 0.002 | 0.000 | 1.000 | 0.500 | 0.000 | **0.532** | 0.476 | **0.504** | **0.467** |
| Average | 0.438 | 0.968 | 0.703 | 0.438 | 0.736 | 0.804 | **0.770** | **0.713** | 0.273 | **0.987** | 0.630 | 0.273 | 0.398 | 0.971 | 0.684 | 0.398 | **0.747** | 0.744 | 0.745 | 0.690 |

*(d)* $N = 100$ and $\pi_1 = 0.30$

| Dataset | None | | | | Thrsh. | | | | OS | | | | TabPFGen | | | | DS | | | |
|---|---|---|---|---|---|---|---|---|---|---|---|---|---|---|---|---|---|---|---|---|
| | Min. | Maj. | Bal. | WCA | Min. | Maj. | Bal. | WCA | Min. | Maj. | Bal. | WCA | Min. | Maj. | Bal. | WCA | Min. | Maj. | Bal. | WCA |
| kr-vs-kp | 0.841 | 0.976 | 0.908 | 0.841 | 0.881 | 0.962 | 0.922 | 0.881 | 0.816 | **0.982** | 0.899 | 0.816 | 0.835 | 0.979 | 0.907 | 0.835 | **0.920** | 0.945 | **0.933** | **0.899** |
| spambase | 0.817 | 0.951 | 0.884 | 0.817 | 0.877 | 0.910 | **0.893** | 0.874 | 0.741 | **0.957** | 0.849 | 0.741 | 0.789 | 0.954 | 0.871 | 0.789 | **0.892** | 0.888 | 0.890 | **0.875** |
| electricity | 0.420 | **0.941** | 0.680 | 0.420 | **0.680** | 0.803 | **0.741** | **0.680** | 0.374 | 0.927 | 0.650 | 0.374 | 0.452 | 0.923 | 0.688 | 0.452 | 0.678 | 0.781 | 0.730 | 0.667 |
| jm1 | 0.254 | **0.899** | 0.576 | 0.254 | 0.574 | 0.680 | **0.627** | **0.574** | 0.283 | 0.861 | 0.572 | 0.283 | 0.308 | 0.865 | 0.587 | 0.308 | **0.576** | 0.678 | 0.627 | 0.566 |
| adult | 0.602 | 0.867 | 0.734 | 0.602 | 0.794 | 0.763 | **0.778** | **0.733** | 0.505 | **0.896** | 0.700 | 0.505 | 0.596 | 0.865 | 0.731 | 0.596 | **0.836** | 0.710 | 0.773 | 0.705 |
| Bioresponse | 0.260 | 0.921 | 0.591 | 0.260 | **0.685** | 0.658 | 0.671 | 0.620 | 0.224 | **0.925** | 0.574 | 0.224 | 0.297 | 0.901 | 0.599 | 0.297 | 0.675 | 0.678 | **0.676** | **0.644** |
| phoneme | 0.605 | 0.878 | 0.741 | 0.605 | **0.821** | 0.723 | **0.772** | 0.723 | 0.489 | **0.902** | 0.695 | 0.489 | 0.533 | 0.892 | 0.712 | 0.533 | 0.786 | 0.741 | 0.764 | **0.731** |
| nomao | 0.861 | 0.928 | 0.894 | 0.861 | 0.907 | 0.893 | **0.900** | **0.882** | 0.778 | **0.953** | 0.866 | 0.778 | 0.793 | 0.950 | 0.871 | 0.793 | 0.891 | 0.892 | 0.891 | 0.875 |
| PhishingWebsites | 0.824 | **0.970** | 0.897 | 0.824 | 0.881 | 0.932 | 0.906 | 0.877 | 0.810 | 0.966 | 0.888 | 0.810 | 0.818 | 0.969 | 0.894 | 0.818 | 0.898 | 0.931 | **0.915** | **0.893** |
| bank-marketing | 0.573 | 0.896 | 0.734 | 0.573 | 0.777 | 0.783 | 0.780 | 0.740 | 0.372 | **0.931** | 0.651 | 0.372 | 0.520 | 0.908 | 0.714 | 0.520 | **0.781** | 0.780 | **0.781** | **0.744** |
| numerai28.6 | 0.012 | **0.991** | 0.501 | 0.012 | 0.414 | 0.599 | **0.506** | 0.414 | 0.031 | 0.972 | 0.501 | 0.031 | 0.034 | 0.968 | 0.501 | 0.034 | **0.510** | 0.500 | 0.505 | **0.461** |
| Average | 0.552 | 0.929 | 0.740 | 0.551 | 0.754 | 0.791 | **0.773** | 0.727 | 0.493 | **0.934** | 0.713 | 0.493 | 0.543 | 0.925 | 0.734 | 0.543 | **0.768** | 0.775 | 0.771 | **0.733** |

*Table 3.* Results by Dataset and Correction Method (continued)

### (e) N = 500 and π₁ = 0.05

| Dataset | None | | | | Thrsh. | | | | OS | | | | TabPFGen | | | | DS | | | |
|---|---|---|---|---|---|---|---|---|---|---|---|---|---|---|---|---|---|---|---|---|
| | Min. | Maj. | Bal. | WCA | Min. | Maj. | Bal. | WCA | Min. | Maj. | Bal. | WCA | Min. | Maj. | Bal. | WCA | Min. | Maj. | Bal. | WCA |
| kr-vs-kp | 0.807 | 0.997 | 0.902 | 0.807 | **0.924** | 0.968 | **0.946** | **0.921** | 0.117 | **1.000** | 0.559 | 0.117 | 0.807 | 0.997 | 0.902 | 0.807 | 0.903 | 0.906 | 0.905 | 0.868 |
| spambase | 0.609 | 0.991 | 0.800 | 0.609 | **0.884** | 0.940 | **0.912** | **0.880** | 0.071 | **1.000** | 0.535 | 0.071 | 0.299 | 0.997 | 0.648 | 0.299 | 0.873 | 0.904 | 0.888 | 0.859 |
| electricity | 0.083 | 0.997 | 0.540 | 0.083 | 0.651 | 0.840 | **0.745** | 0.651 | 0.005 | **1.000** | 0.502 | 0.005 | 0.075 | 0.998 | 0.536 | 0.075 | **0.681** | 0.750 | 0.716 | **0.665** |
| jm1 | 0.001 | **1.000** | 0.500 | 0.001 | 0.493 | 0.763 | **0.628** | 0.492 | 0.006 | 0.994 | 0.500 | 0.006 | 0.003 | 0.999 | 0.501 | 0.003 | **0.588** | 0.644 | 0.616 | **0.533** |
| adult | 0.142 | 0.996 | 0.569 | 0.142 | 0.769 | 0.804 | **0.787** | **0.723** | 0.006 | **1.000** | 0.503 | 0.006 | 0.136 | 0.996 | 0.566 | 0.136 | **0.783** | 0.714 | 0.749 | 0.698 |
| Bioresponse | 0.025 | 0.997 | 0.511 | 0.025 | **0.702** | 0.680 | **0.691** | **0.639** | 0.002 | **1.000** | 0.501 | 0.002 | 0.025 | 0.997 | 0.511 | 0.025 | 0.652 | 0.653 | 0.652 | 0.597 |
| phoneme | 0.055 | 0.997 | 0.526 | 0.055 | **0.827** | 0.778 | **0.802** | **0.762** | 0.012 | **0.999** | 0.505 | 0.012 | 0.043 | 0.998 | 0.520 | 0.043 | 0.806 | 0.723 | 0.765 | 0.714 |
| nomao | 0.620 | 0.991 | 0.805 | 0.620 | 0.900 | 0.929 | **0.915** | **0.892** | 0.091 | **1.000** | 0.546 | 0.091 | 0.458 | 0.996 | 0.727 | 0.458 | **0.901** | 0.878 | 0.890 | 0.863 |
| PhishingWebsites | 0.702 | 0.997 | 0.850 | 0.702 | 0.871 | 0.954 | **0.912** | **0.871** | 0.128 | **1.000** | 0.564 | 0.128 | 0.694 | 0.997 | 0.845 | 0.694 | **0.892** | 0.909 | 0.900 | 0.868 |
| bank-marketing | 0.085 | 0.995 | 0.540 | 0.085 | 0.750 | 0.839 | **0.795** | **0.746** | 0.000 | **1.000** | 0.500 | 0.000 | 0.083 | 0.995 | 0.539 | 0.083 | **0.763** | 0.780 | 0.771 | 0.739 |
| numerai28.6 | 0.000 | **1.000** | 0.500 | 0.000 | 0.197 | 0.812 | **0.504** | 0.197 | 0.000 | 1.000 | 0.500 | 0.000 | 0.000 | 1.000 | 0.500 | 0.000 | **0.513** | 0.494 | 0.503 | **0.451** |
| Average | 0.284 | 0.996 | 0.640 | 0.284 | 0.724 | 0.846 | **0.785** | 0.707 | 0.040 | **0.999** | 0.520 | 0.040 | 0.238 | 0.997 | 0.618 | 0.238 | **0.759** | 0.760 | 0.760 | **0.714** |

### (f) N = 500 and π₁ = 0.10

| Dataset | None | | | | Thrsh. | | | | OS | | | | TabPFGen | | | | DS | | | |
|---|---|---|---|---|---|---|---|---|---|---|---|---|---|---|---|---|---|---|---|---|
| | Min. | Maj. | Bal. | WCA | Min. | Maj. | Bal. | WCA | Min. | Maj. | Bal. | WCA | Min. | Maj. | Bal. | WCA | Min. | Maj. | Bal. | WCA |
| kr-vs-kp | 0.876 | 0.995 | 0.935 | 0.876 | **0.948** | 0.971 | **0.960** | **0.944** | 0.452 | **1.000** | 0.726 | 0.452 | 0.866 | 0.995 | 0.930 | 0.866 | 0.935 | 0.948 | 0.941 | 0.920 |
| spambase | 0.742 | 0.985 | 0.863 | 0.742 | **0.913** | 0.935 | **0.924** | **0.908** | 0.204 | 0.998 | 0.601 | 0.204 | 0.556 | 0.993 | 0.775 | 0.556 | 0.897 | 0.918 | 0.908 | 0.883 |
| electricity | 0.256 | 0.989 | 0.622 | 0.256 | 0.672 | 0.845 | **0.759** | 0.672 | 0.024 | **0.999** | 0.511 | 0.024 | 0.211 | 0.990 | 0.600 | 0.211 | **0.688** | 0.792 | 0.740 | **0.673** |
| jm1 | 0.013 | **0.998** | 0.505 | 0.013 | 0.543 | 0.742 | **0.642** | 0.543 | 0.021 | 0.991 | 0.506 | 0.021 | 0.027 | 0.996 | 0.511 | 0.027 | **0.584** | 0.683 | 0.633 | **0.562** |
| adult | 0.267 | 0.988 | 0.627 | 0.267 | 0.802 | 0.802 | **0.802** | 0.766 | 0.019 | **0.999** | 0.509 | 0.019 | 0.267 | 0.988 | 0.627 | 0.267 | **0.841** | 0.736 | 0.789 | 0.730 |
| Bioresponse | 0.077 | 0.991 | 0.534 | 0.077 | **0.751** | 0.675 | **0.713** | **0.668** | 0.008 | **0.999** | 0.503 | 0.008 | 0.079 | 0.991 | 0.535 | 0.079 | 0.696 | 0.682 | 0.689 | 0.647 |
| phoneme | 0.228 | 0.985 | 0.607 | 0.228 | **0.854** | 0.788 | **0.821** | **0.785** | 0.043 | **0.998** | 0.520 | 0.043 | 0.127 | 0.990 | 0.559 | 0.127 | 0.836 | 0.747 | 0.792 | 0.743 |
| nomao | 0.759 | 0.982 | 0.871 | 0.759 | 0.913 | 0.929 | **0.921** | **0.905** | 0.222 | **1.000** | 0.611 | 0.222 | 0.634 | 0.992 | 0.813 | 0.634 | **0.923** | 0.897 | 0.910 | 0.890 |
| PhishingWebsites | 0.762 | 0.993 | 0.878 | 0.762 | 0.900 | 0.949 | **0.924** | **0.898** | 0.413 | 0.999 | 0.706 | 0.413 | 0.762 | 0.993 | 0.877 | 0.762 | **0.904** | 0.922 | 0.913 | 0.892 |
| bank-marketing | 0.246 | 0.981 | 0.613 | 0.246 | 0.795 | 0.828 | **0.811** | **0.782** | 0.002 | **1.000** | 0.501 | 0.002 | 0.242 | 0.981 | 0.611 | 0.242 | **0.802** | 0.787 | 0.795 | 0.761 |
| numerai28.6 | 0.000 | **1.000** | 0.500 | 0.000 | 0.205 | 0.801 | 0.503 | 0.205 | 0.000 | 1.000 | 0.500 | 0.000 | 0.000 | 1.000 | 0.500 | 0.000 | **0.503** | 0.510 | **0.506** | 0.469 |
| Average | 0.384 | 0.990 | 0.687 | 0.384 | 0.754 | 0.842 | **0.798** | 0.734 | 0.128 | **0.998** | 0.563 | 0.128 | 0.343 | 0.992 | 0.667 | 0.343 | **0.783** | 0.784 | 0.783 | **0.743** |

### (g) N = 500 and π₁ = 0.20

| Dataset | None | | | | Thrsh. | | | | OS | | | | TabPFGen | | | | DS | | | |
|---|---|---|---|---|---|---|---|---|---|---|---|---|---|---|---|---|---|---|---|---|
| | Min. | Maj. | Bal. | WCA | Min. | Maj. | Bal. | WCA | Min. | Maj. | Bal. | WCA | Min. | Maj. | Bal. | WCA | Min. | Maj. | Bal. | WCA |
| kr-vs-kp | 0.936 | 0.992 | 0.964 | 0.936 | **0.973** | 0.974 | **0.973** | **0.966** | 0.853 | **0.998** | 0.925 | 0.853 | 0.930 | 0.992 | 0.961 | 0.930 | 0.956 | 0.952 | 0.954 | 0.941 |
| spambase | 0.839 | 0.976 | 0.907 | 0.839 | **0.927** | 0.941 | **0.934** | **0.923** | 0.558 | **0.991** | 0.775 | 0.558 | 0.759 | 0.984 | 0.871 | 0.759 | 0.916 | 0.934 | 0.925 | 0.910 |
| electricity | 0.418 | 0.969 | 0.693 | 0.418 | 0.690 | 0.838 | **0.764** | 0.689 | 0.185 | **0.988** | 0.586 | 0.185 | 0.408 | 0.968 | 0.688 | 0.408 | **0.699** | 0.800 | 0.749 | **0.694** |
| jm1 | 0.105 | **0.977** | 0.541 | 0.105 | 0.548 | 0.745 | **0.647** | 0.548 | 0.094 | 0.961 | 0.528 | 0.094 | 0.130 | 0.971 | 0.551 | 0.130 | **0.596** | 0.689 | 0.642 | **0.563** |
| adult | 0.497 | 0.951 | 0.724 | 0.497 | 0.830 | 0.783 | **0.807** | 0.775 | 0.150 | **0.990** | 0.570 | 0.150 | 0.494 | 0.950 | 0.722 | 0.494 | **0.857** | 0.748 | 0.803 | 0.747 |
| Bioresponse | 0.292 | 0.956 | 0.624 | 0.292 | **0.763** | 0.701 | **0.732** | 0.699 | 0.071 | **0.993** | 0.532 | 0.071 | 0.290 | 0.958 | 0.624 | 0.290 | 0.735 | 0.686 | 0.711 | 0.681 |
| phoneme | 0.557 | 0.942 | 0.749 | 0.557 | **0.862** | 0.799 | **0.831** | **0.796** | 0.257 | **0.979** | 0.618 | 0.257 | 0.343 | 0.969 | 0.656 | 0.343 | 0.857 | 0.773 | 0.815 | 0.769 |
| nomao | 0.847 | 0.972 | 0.910 | 0.847 | **0.935** | 0.929 | **0.932** | **0.920** | 0.592 | **0.995** | 0.794 | 0.592 | 0.805 | 0.981 | 0.893 | 0.805 | 0.930 | 0.913 | 0.922 | 0.910 |
| PhishingWebsites | 0.832 | 0.984 | 0.908 | 0.832 | **0.915** | 0.947 | **0.931** | **0.914** | 0.713 | 0.994 | 0.853 | 0.713 | 0.832 | 0.984 | 0.908 | 0.832 | 0.913 | 0.940 | 0.927 | 0.909 |
| bank-marketing | 0.530 | 0.942 | 0.736 | 0.530 | **0.834** | 0.820 | **0.827** | **0.811** | 0.060 | **0.995** | 0.527 | 0.060 | 0.488 | 0.948 | 0.718 | 0.488 | 0.832 | 0.795 | 0.814 | 0.788 |
| numerai28.6 | 0.000 | **1.000** | 0.500 | 0.000 | 0.154 | 0.853 | **0.504** | 0.154 | 0.002 | 0.998 | 0.500 | 0.002 | 0.000 | 1.000 | 0.500 | 0.000 | **0.487** | 0.513 | 0.500 | **0.441** |
| Average | 0.532 | 0.969 | 0.751 | 0.532 | 0.766 | 0.848 | **0.807** | 0.745 | 0.321 | **0.989** | 0.655 | 0.321 | 0.498 | 0.973 | 0.736 | 0.498 | **0.798** | 0.795 | 0.796 | **0.759** |

### (h) N = 500 and π₁ = 0.30

| Dataset | None | | | | Thrsh. | | | | OS | | | | TabPFGen | | | | DS | | | |
|---|---|---|---|---|---|---|---|---|---|---|---|---|---|---|---|---|---|---|---|---|
| | Min. | Maj. | Bal. | WCA | Min. | Maj. | Bal. | WCA | Min. | Maj. | Bal. | WCA | Min. | Maj. | Bal. | WCA | Min. | Maj. | Bal. | WCA |
| kr-vs-kp | 0.960 | 0.988 | 0.974 | 0.959 | **0.977** | 0.979 | **0.978** | **0.968** | 0.940 | **0.993** | 0.967 | 0.940 | 0.957 | 0.989 | 0.973 | 0.956 | 0.971 | 0.962 | 0.966 | 0.953 |
| spambase | 0.892 | 0.968 | 0.930 | 0.892 | **0.931** | 0.946 | **0.938** | **0.927** | 0.825 | **0.978** | 0.901 | 0.825 | 0.874 | 0.970 | 0.922 | 0.874 | 0.927 | 0.938 | 0.933 | 0.921 |
| electricity | 0.532 | **0.937** | 0.735 | 0.532 | **0.719** | 0.830 | **0.775** | **0.719** | 0.458 | 0.936 | 0.697 | 0.458 | 0.544 | 0.932 | 0.738 | 0.544 | 0.719 | 0.803 | 0.761 | 0.710 |
| jm1 | 0.255 | **0.926** | 0.591 | 0.255 | 0.575 | 0.724 | 0.650 | 0.574 | 0.277 | 0.885 | 0.581 | 0.277 | 0.284 | 0.910 | 0.597 | 0.284 | **0.622** | 0.683 | **0.653** | **0.608** |
| adult | 0.661 | 0.907 | 0.784 | 0.661 | **0.851** | 0.781 | **0.816** | **0.781** | 0.531 | 0.926 | 0.728 | 0.531 | 0.660 | 0.907 | 0.783 | 0.660 | 0.850 | 0.765 | 0.807 | 0.764 |
| Bioresponse | 0.495 | 0.898 | 0.697 | 0.495 | **0.758** | 0.711 | **0.735** | 0.705 | 0.298 | **0.951** | 0.625 | 0.298 | 0.460 | 0.908 | 0.684 | 0.460 | 0.733 | 0.710 | 0.721 | 0.698 |
| phoneme | 0.726 | 0.892 | 0.809 | 0.726 | **0.879** | 0.792 | **0.836** | **0.791** | 0.612 | **0.923** | 0.767 | 0.612 | 0.647 | 0.917 | 0.782 | 0.647 | 0.874 | 0.775 | 0.824 | 0.773 |
| nomao | 0.891 | 0.960 | 0.926 | 0.891 | 0.934 | 0.930 | **0.932** | **0.923** | 0.826 | 0.978 | 0.902 | 0.826 | 0.877 | 0.967 | 0.922 | 0.877 | **0.937** | 0.918 | 0.927 | 0.913 |
| PhishingWebsites | 0.879 | 0.969 | 0.924 | 0.879 | **0.920** | 0.942 | **0.931** | **0.915** | 0.839 | **0.981** | 0.910 | 0.839 | 0.879 | 0.969 | 0.924 | 0.879 | 0.918 | 0.942 | 0.930 | 0.915 |
| bank-marketing | 0.689 | 0.902 | 0.795 | 0.689 | **0.850** | 0.813 | **0.831** | **0.806** | 0.446 | **0.946** | 0.696 | 0.446 | 0.680 | 0.903 | 0.792 | 0.680 | 0.835 | 0.814 | 0.824 | 0.804 |
| numerai28.6 | 0.000 | **1.000** | 0.500 | 0.000 | 0.228 | 0.785 | **0.507** | 0.228 | 0.021 | 0.979 | 0.500 | 0.021 | 0.000 | 1.000 | 0.500 | 0.000 | **0.507** | 0.498 | 0.502 | **0.447** |
| Average | 0.635 | 0.941 | 0.788 | 0.634 | 0.784 | 0.839 | **0.812** | 0.758 | 0.552 | **0.952** | 0.752 | 0.552 | 0.624 | 0.943 | 0.783 | 0.624 | **0.808** | 0.801 | 0.804 | **0.773** |

*Table 3.* Results by Dataset and Correction Method (continued)

*(i) N = 1000 and $\pi_1 = 0.05$*

| Dataset | None | | | | Thrsh. | | | | OS | | | | TabPFGen | | | | DS | | | |
|---|---|---|---|---|---|---|---|---|---|---|---|---|---|---|---|---|---|---|---|---|
| | Min. | Maj. | Bal. | WCA | Min. | Maj. | Bal. | WCA | Min. | Maj. | Bal. | WCA | Min. | Maj. | Bal. | WCA | Min. | Maj. | Bal. | WCA |
| kr-vs-kp | 0.872 | 0.999 | 0.935 | 0.872 | **0.947** | 0.984 | **0.966** | **0.945** | 0.276 | **1.000** | 0.638 | 0.276 | 0.860 | 0.999 | 0.930 | 0.860 | 0.935 | 0.948 | 0.941 | 0.920 |
| spambase | 0.672 | 0.991 | 0.832 | 0.672 | **0.919** | 0.941 | **0.930** | **0.915** | 0.074 | 0.999 | 0.536 | 0.074 | 0.354 | 0.997 | 0.675 | 0.354 | 0.897 | 0.918 | 0.908 | 0.883 |
| electricity | 0.132 | 0.997 | 0.564 | 0.132 | 0.674 | 0.860 | **0.767** | **0.674** | 0.005 | **1.000** | 0.502 | 0.005 | 0.117 | 0.997 | 0.557 | 0.117 | **0.688** | 0.792 | 0.740 | 0.673 |
| jm1 | 0.001 | **1.000** | 0.500 | 0.001 | 0.454 | 0.806 | 0.630 | 0.454 | 0.010 | 0.994 | 0.502 | 0.010 | 0.003 | 1.000 | 0.501 | 0.003 | **0.584** | 0.683 | **0.633** | **0.562** |
| adult | 0.138 | 0.996 | 0.567 | 0.138 | 0.788 | 0.804 | **0.796** | **0.760** | 0.004 | **1.000** | 0.502 | 0.004 | 0.133 | 0.996 | 0.565 | 0.133 | **0.841** | 0.736 | 0.789 | 0.730 |
| Bioresponse | 0.041 | 0.997 | 0.519 | 0.041 | **0.826** | 0.547 | 0.687 | 0.547 | 0.002 | **1.000** | 0.501 | 0.002 | 0.042 | 0.997 | 0.520 | 0.042 | 0.696 | 0.682 | **0.689** | **0.647** |
| phoneme | 0.101 | 0.997 | 0.549 | 0.101 | **0.856** | 0.791 | **0.824** | **0.785** | 0.022 | **1.000** | 0.511 | 0.022 | 0.080 | 0.998 | 0.539 | 0.080 | 0.836 | 0.747 | 0.792 | 0.743 |
| nomao | 0.677 | 0.989 | 0.833 | 0.677 | 0.896 | 0.935 | **0.916** | **0.896** | 0.145 | **1.000** | 0.572 | 0.145 | 0.546 | 0.996 | 0.771 | 0.546 | **0.923** | 0.897 | 0.910 | 0.890 |
| PhishingWebsites | 0.709 | 0.999 | 0.854 | 0.709 | 0.880 | 0.963 | **0.921** | **0.880** | 0.195 | 0.999 | 0.597 | 0.195 | 0.710 | 0.999 | 0.855 | 0.710 | **0.904** | 0.922 | 0.913 | **0.892** |
| bank-marketing | 0.081 | 0.996 | 0.538 | 0.081 | **0.823** | 0.834 | **0.828** | **0.810** | 0.000 | **1.000** | 0.500 | 0.000 | 0.080 | 0.996 | 0.538 | 0.080 | 0.802 | 0.787 | 0.795 | 0.761 |
| numerai28.6 | 0.000 | **1.000** | 0.500 | 0.000 | 0.194 | 0.818 | 0.506 | 0.194 | 0.000 | 1.000 | 0.500 | 0.000 | 0.000 | 1.000 | 0.500 | 0.000 | **0.503** | 0.510 | **0.506** | 0.469 |
| Average | 0.311 | 0.997 | 0.654 | 0.311 | 0.751 | 0.844 | **0.797** | 0.715 | 0.067 | **0.999** | 0.533 | 0.067 | 0.266 | 0.998 | 0.632 | 0.266 | **0.783** | 0.784 | 0.783 | **0.743** |

*(j) N = 1000 and $\pi_1 = 0.10$*

| Dataset | None | | | | Thrsh. | | | | OS | | | | TabPFGen | | | | DS | | | |
|---|---|---|---|---|---|---|---|---|---|---|---|---|---|---|---|---|---|---|---|---|
| | Min. | Maj. | Bal. | WCA | Min. | Maj. | Bal. | WCA | Min. | Maj. | Bal. | WCA | Min. | Maj. | Bal. | WCA | Min. | Maj. | Bal. | WCA |
| kr-vs-kp | 0.924 | 0.998 | 0.961 | 0.924 | **0.977** | 0.988 | **0.982** | **0.974** | 0.634 | **1.000** | 0.817 | 0.634 | 0.919 | 0.998 | 0.959 | 0.919 | 0.956 | 0.952 | 0.954 | 0.941 |
| spambase | 0.791 | 0.987 | 0.889 | 0.791 | **0.930** | 0.945 | **0.938** | **0.926** | 0.244 | 0.999 | 0.621 | 0.244 | 0.595 | 0.995 | 0.795 | 0.595 | 0.916 | 0.934 | 0.925 | 0.910 |
| electricity | 0.296 | 0.989 | 0.642 | 0.296 | 0.686 | 0.861 | **0.774** | 0.686 | 0.020 | **0.999** | 0.510 | 0.020 | 0.218 | 0.993 | 0.605 | 0.218 | **0.699** | 0.800 | 0.749 | **0.694** |
| jm1 | 0.009 | **0.999** | 0.504 | 0.009 | 0.526 | 0.764 | **0.645** | 0.526 | 0.027 | 0.988 | 0.508 | 0.027 | 0.019 | 0.998 | 0.508 | 0.019 | **0.596** | 0.689 | 0.642 | **0.563** |
| adult | 0.279 | 0.986 | 0.632 | 0.279 | 0.822 | 0.798 | **0.810** | **0.778** | 0.017 | **0.999** | 0.508 | 0.017 | 0.276 | 0.985 | 0.630 | 0.276 | **0.857** | 0.748 | 0.803 | 0.747 |
| Bioresponse | 0.166 | 0.990 | 0.578 | 0.166 | **0.816** | 0.632 | **0.724** | 0.632 | 0.012 | **0.999** | 0.505 | 0.012 | 0.164 | 0.991 | 0.577 | 0.164 | 0.735 | 0.686 | 0.711 | **0.681** |
| phoneme | 0.310 | 0.983 | 0.647 | 0.310 | **0.861** | 0.803 | **0.832** | **0.801** | 0.066 | 0.998 | 0.532 | 0.066 | 0.158 | 0.993 | 0.576 | 0.158 | 0.857 | 0.773 | 0.815 | 0.769 |
| nomao | 0.763 | 0.985 | 0.874 | 0.763 | 0.916 | 0.932 | **0.924** | **0.915** | 0.268 | 0.999 | 0.634 | 0.268 | 0.689 | 0.990 | 0.839 | 0.689 | **0.930** | 0.913 | 0.922 | 0.910 |
| PhishingWebsites | 0.775 | 0.994 | 0.885 | 0.775 | **0.915** | 0.951 | **0.933** | **0.915** | 0.503 | 0.998 | 0.750 | 0.503 | 0.776 | 0.994 | 0.885 | 0.776 | 0.913 | 0.940 | 0.927 | 0.909 |
| bank-marketing | 0.263 | 0.985 | 0.624 | 0.263 | **0.852** | 0.828 | **0.840** | **0.823** | 0.001 | **1.000** | 0.500 | 0.001 | 0.255 | 0.985 | 0.620 | 0.255 | 0.832 | 0.795 | 0.814 | 0.788 |
| numerai28.6 | 0.000 | **1.000** | 0.500 | 0.000 | 0.127 | 0.873 | **0.500** | 0.127 | 0.000 | 1.000 | 0.500 | 0.000 | 0.000 | 1.000 | 0.500 | 0.000 | **0.487** | 0.513 | 0.500 | 0.441 |
| Average | 0.416 | 0.991 | 0.703 | 0.416 | 0.766 | 0.852 | **0.809** | 0.737 | 0.163 | **0.998** | 0.580 | 0.163 | 0.370 | 0.993 | 0.681 | 0.370 | **0.798** | 0.795 | 0.796 | **0.759** |

*(k) N = 1000 and $\pi_1 = 0.20$*

| Dataset | None | | | | Thrsh. | | | | OS | | | | TabPFGen | | | | DS | | | |
|---|---|---|---|---|---|---|---|---|---|---|---|---|---|---|---|---|---|---|---|---|
| | Min. | Maj. | Bal. | WCA | Min. | Maj. | Bal. | WCA | Min. | Maj. | Bal. | WCA | Min. | Maj. | Bal. | WCA | Min. | Maj. | Bal. | WCA |
| kr-vs-kp | 0.976 | 0.996 | 0.986 | 0.976 | **0.991** | 0.991 | **0.991** | **0.988** | 0.928 | **0.998** | 0.963 | 0.928 | 0.973 | 0.995 | 0.984 | 0.973 | 0.980 | 0.966 | 0.973 | 0.958 |
| spambase | 0.876 | 0.976 | 0.926 | 0.876 | **0.941** | 0.943 | **0.942** | **0.931** | 0.624 | **0.993** | 0.809 | 0.624 | 0.796 | 0.986 | 0.891 | 0.796 | 0.935 | 0.936 | 0.935 | 0.928 |
| electricity | 0.462 | 0.970 | 0.716 | 0.462 | 0.729 | 0.844 | **0.787** | 0.729 | 0.200 | **0.990** | 0.595 | 0.200 | 0.437 | 0.970 | 0.703 | 0.437 | **0.735** | 0.807 | 0.771 | **0.733** |
| jm1 | 0.102 | **0.975** | 0.539 | 0.102 | 0.562 | 0.735 | 0.649 | 0.562 | 0.109 | 0.964 | 0.537 | 0.109 | 0.126 | 0.966 | 0.546 | 0.126 | **0.610** | 0.694 | **0.652** | **0.609** |
| adult | 0.534 | 0.947 | 0.741 | 0.534 | **0.844** | 0.792 | **0.818** | **0.791** | 0.181 | **0.988** | 0.584 | 0.181 | 0.533 | 0.947 | 0.740 | 0.533 | 0.838 | 0.782 | 0.810 | 0.779 |
| Bioresponse | 0.397 | 0.953 | 0.675 | 0.397 | **0.809** | 0.686 | **0.747** | 0.686 | 0.098 | **0.991** | 0.545 | 0.098 | 0.387 | 0.953 | 0.670 | 0.387 | 0.744 | 0.733 | 0.738 | **0.721** |
| phoneme | 0.608 | 0.945 | 0.777 | 0.608 | 0.882 | 0.811 | **0.847** | **0.811** | 0.334 | **0.979** | 0.657 | 0.334 | 0.413 | 0.969 | 0.691 | 0.413 | **0.888** | 0.779 | 0.833 | 0.779 |
| nomao | 0.861 | 0.972 | 0.916 | 0.861 | **0.938** | 0.929 | **0.934** | **0.926** | 0.618 | **0.993** | 0.806 | 0.618 | 0.838 | 0.978 | 0.908 | 0.838 | 0.928 | 0.930 | 0.929 | 0.917 |
| PhishingWebsites | 0.852 | 0.984 | 0.918 | 0.852 | **0.928** | 0.953 | **0.940** | **0.928** | 0.742 | **0.993** | 0.868 | 0.742 | 0.852 | 0.984 | 0.918 | 0.852 | 0.922 | 0.936 | 0.929 | 0.912 |
| bank-marketing | 0.590 | 0.942 | 0.766 | 0.590 | **0.879** | 0.820 | **0.849** | **0.820** | 0.051 | **0.997** | 0.524 | 0.051 | 0.531 | 0.946 | 0.738 | 0.531 | 0.854 | 0.810 | 0.832 | 0.810 |
| numerai28.6 | 0.000 | **1.000** | 0.500 | 0.000 | 0.106 | 0.902 | **0.504** | 0.106 | 0.002 | 0.997 | 0.500 | 0.002 | 0.000 | 1.000 | 0.500 | 0.000 | **0.531** | 0.472 | 0.502 | 0.427 |
| Average | 0.569 | 0.969 | 0.769 | 0.569 | 0.783 | 0.855 | **0.819** | 0.753 | 0.354 | **0.989** | 0.671 | 0.354 | 0.535 | 0.972 | 0.754 | 0.535 | **0.815** | 0.804 | 0.810 | **0.779** |

*(l) N = 1000 and $\pi_1 = 0.30$*

| Dataset | None | | | | Thrsh. | | | | OS | | | | TabPFGen | | | | DS | | | |
|---|---|---|---|---|---|---|---|---|---|---|---|---|---|---|---|---|---|---|---|---|
| | Min. | Maj. | Bal. | WCA | Min. | Maj. | Bal. | WCA | Min. | Maj. | Bal. | WCA | Min. | Maj. | Bal. | WCA | Min. | Maj. | Bal. | WCA |
| kr-vs-kp | 0.980 | 0.994 | 0.987 | 0.979 | **0.988** | 0.989 | **0.989** | **0.984** | 0.972 | **0.996** | 0.984 | 0.972 | 0.979 | 0.994 | 0.987 | 0.978 | 0.987 | 0.981 | 0.984 | 0.977 |
| spambase | 0.904 | 0.970 | 0.937 | 0.904 | **0.940** | 0.947 | **0.944** | **0.938** | 0.840 | **0.980** | 0.910 | 0.840 | 0.885 | 0.976 | 0.930 | 0.885 | 0.937 | 0.942 | 0.940 | 0.935 |
| electricity | 0.582 | 0.935 | 0.758 | 0.582 | 0.749 | 0.835 | **0.792** | 0.746 | 0.508 | **0.947** | 0.728 | 0.508 | 0.582 | 0.934 | 0.758 | 0.582 | **0.758** | 0.815 | 0.787 | **0.750** |
| jm1 | 0.304 | **0.906** | 0.605 | 0.304 | 0.577 | 0.723 | 0.650 | 0.575 | 0.299 | 0.883 | 0.591 | 0.299 | 0.310 | 0.900 | 0.605 | 0.310 | **0.620** | 0.693 | **0.657** | **0.611** |
| adult | 0.678 | 0.903 | 0.791 | 0.678 | 0.847 | 0.798 | **0.823** | **0.798** | 0.577 | **0.919** | 0.748 | 0.577 | 0.673 | 0.903 | 0.788 | 0.673 | **0.862** | 0.770 | 0.816 | 0.770 |
| Bioresponse | 0.558 | 0.901 | 0.730 | 0.558 | **0.793** | 0.730 | **0.762** | 0.727 | 0.362 | **0.957** | 0.659 | 0.362 | 0.519 | 0.921 | 0.720 | 0.519 | 0.766 | 0.746 | 0.756 | **0.739** |
| phoneme | 0.767 | 0.897 | 0.832 | 0.767 | 0.891 | 0.820 | **0.856** | **0.820** | 0.688 | **0.930** | 0.809 | 0.688 | 0.704 | 0.925 | 0.815 | 0.704 | **0.893** | 0.781 | 0.837 | 0.781 |
| nomao | 0.900 | 0.958 | 0.929 | 0.900 | **0.945** | 0.932 | **0.939** | **0.931** | 0.848 | **0.976** | 0.912 | 0.848 | 0.892 | 0.965 | 0.928 | 0.892 | 0.941 | 0.923 | 0.932 | 0.919 |
| PhishingWebsites | 0.890 | 0.977 | 0.934 | 0.890 | **0.936** | 0.952 | **0.944** | **0.933** | 0.860 | **0.979** | 0.920 | 0.860 | 0.890 | 0.977 | 0.934 | 0.890 | 0.929 | 0.948 | 0.938 | 0.926 |
| bank-marketing | 0.735 | 0.899 | 0.817 | 0.735 | **0.878** | 0.823 | **0.851** | **0.823** | 0.500 | **0.946** | 0.723 | 0.500 | 0.720 | 0.898 | 0.809 | 0.720 | 0.862 | 0.816 | 0.839 | 0.813 |
| numerai28.6 | 0.000 | **1.000** | 0.500 | 0.000 | 0.256 | 0.755 | **0.505** | 0.256 | 0.030 | 0.978 | 0.504 | 0.030 | 0.000 | 1.000 | 0.500 | 0.000 | **0.478** | 0.527 | 0.503 | 0.448 |
| Average | 0.664 | 0.940 | 0.802 | 0.663 | 0.800 | 0.846 | **0.823** | 0.776 | 0.589 | **0.954** | 0.772 | 0.589 | 0.650 | 0.945 | 0.798 | 0.650 | **0.821** | 0.813 | 0.817 | **0.788** |

