# OpenReview forum: "Correcting Class Imbalance in Prior-Data Fitted Networks for Tabular Classification"
_ICML.cc/2026/Workshop/FMSD — FMSD @ ICML 2026 Poster_

### Official Review · Reviewer_hmw7 · 2026-05-15
**A Practical Empirical Study on Class-Imbalance Correction for TabPFN**

**Rating:** 6
**Confidence:** 4

**Review:**

This paper looks at how class imbalance affects PFN for tabular classification, mainly using TabPFN as the representative model. Since PFNs make predictions through in-context learning and do not update their weights on a new task, some standard imbalance-handling methods, such as loss reweighting, are not directly applicable. The authors therefore compare several simple alternatives, including threshold adjustment, downsampling, oversampling, and synthetic upsampling. Their main finding is that thresholding and downsampling work surprisingly well for improving minority-class and worst-class accuracy, while oversampling and synthetic upsampling are less helpful.

Strengths:

The paper addresses a practical issue that is very relevant for using tabular foundation models in real applications. Class imbalance is common in tabular problems, and the paper makes a good case that this issue should be studied specifically for PFNs rather than assuming that standard recipes will behave the same way.


Areas for Improvement:

The main weakness is that the contribution is mostly empirical and the methods themselves are quite standard. Thresholding and downsampling are classical techniques, so the novelty mainly comes from showing that they are effective in the PFN setting. This is useful, but it would be stronger if the paper provided a deeper explanation of why these methods work particularly well for PFNs.

The experimental scope is also somewhat limited. The study focuses on binary classification and uses controlled class imbalance constructed from a subset of OpenML-CC18. This makes the experiments clean, but it also leaves open the question of how well the conclusions transfer to naturally imbalanced datasets or multi-class tabular problems. Since many real structured-data applications involve natural imbalance and more than two classes, this would be an important direction to address.

Detailed Comments:

1. The paper should more clearly distinguish between correcting for an imbalanced context set and optimizing for a particular deployment distribution. In the current experiments, the test set is balanced, so thresholding at the minority prior is closely tied to optimizing balanced accuracy. It would be useful to explain what a practitioner should do when the test distribution is also imbalanced.

2. The calibration plots are interesting, but the explanation is still mostly empirical. I would like to see more discussion of why PFNs show this kind of majority-class bias and why the simple threshold rule works so consistently.

3. The paper would be stronger with multi-class experiments, or at least a more concrete discussion of how the proposed correction strategies would need to change in the multi-class case.

4. The comparison with synthetic upsampling is useful, but the paper should clarify more carefully how the synthetic samples are generated and whether the generator suffers from the same lack of minority-class information as the classifier.

5. The point about downsampling reducing inference cost is important and practical. It would be helpful to report runtime or memory usage in addition to accuracy, since this is one of the main advantages of downsampling for TabPFN.

6. Some additional calibration or post-hoc correction baselines, such as temperature scaling, Platt scaling, isotonic regression, or prior correction with a validation set, would make the comparison more complete.

---

### Official Review · Reviewer_R6uh · 2026-05-21
**Nice study on imbalanced ML for TabPFN**

**Rating:** 7
**Confidence:** 3

**Review:**

The paper "Correcting Class Imbalance in Prior-Data Fitted Networks for Tabular  Classification" evaluates how TabPFN performs for imbalanced data and tests several simple heuristics to improve classification performance in these scenarios. To better understand this, the paper first does a controlled study of how imbalanced classes affect calibration. Then, they derive a simple heuristic strategy to adapt the threshold. In final experiments, the threshold and standard undersampling both improve the performance for TabPFN for imbalanced data.

## Strengths
* Simple idea to improve performance for imbalanced data.
* Understudied topic. To the best of my knowledge, imbalanced data for TabPFN has not yet been tackled.

## Areas for Improvements

* The referencing of related work is rather sparse. There are plenty of works on oversampling and downsampling from the pre-TabPFN era, and also plenty of papers on synthetic upsampling. These should be taken into account, as they can provide valuable insights (and as they realized the issues with these methods, which are described in the paper, but not attributed correctly).
* The writing in Section 2 is very repetitive. Maybe 2.1 and 2.2 can be streamlined to repeat the introduction less.
* It could be interesting to also report an aggregated metric such as the F1 score. Nonetheless, I really like that the paper reports the raw per-class performances in Figure 4.
* The study should not be described as a "thorough empirical investigation" as mentioned in Line 201 (right). While it contains 11 datasets, I think it should be extended to more imbalanced datasets and I would rather see it as a first step.
* The authors should look into the following two works to see if they are related and should be cited:
    * Effective Prompting for Imbalanced-Class Data Synthesis in Tabular Data Classification via Large Language Models
    * Realistic Evaluation of TabPFN v2 in Open Environments
* The paper should cite the OpenML-CC18 benchmarking suite (Bischl et al., NeurIPS 2021).

### Minor
* Line 28 (right): I think the sentence could be improved slightly by specifying that cross-attention captures the relationship between context and query (and not only attention)
* Line 52 (right): wrong cite command. Should use \citet instead of \citep.
* The calculation of the class ration should be given as an equation.
* Figure text is too small across all figures.
* y-axis label in Figure 2 is likely wrong. The paper discusses balanced accuracy everywhere else.
* The references Hollmann et al. (2022) and Müller et al. (2021) should point to ICLR, not arXiv.

In the future it would be interesting to learn if other tabular foundation models behave different or similar.